# Selenium and Its Compounds in the Treatment of Anxiety and Related Disorders: A Scoping Review of Translational and Clinical Research

Ravi Philip Rajkumar 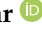

Department of Psychiatry, Jawaharlal Institute of Postgraduate Medical Education and Research (JIPMER), Puducherry 605 006, India; jd0422@jipmer.ac.in; Tel.: +91-413-2296280

**Abstract:** Anxiety disorders are among the most common mental disorders worldwide and often respond incompletely to existing treatments. Selenium, a micronutrient that is a component of several biologically active selenoproteins, is also involved in several aspects of brain functioning and may exert antidepressant and anxiolytic effects through multiple pathways. The current paper is a scoping review of translational, observational, and interventional evidence on the potential role of selenium and its compounds in the management of anxiety and related disorders. Evidence from animal models suggests that this approach may be promising. Though evidence from observational studies in humans is inconsistent and affected by several confounding factors, the available evidence from randomized controlled trials suggests that selenium supplementation may be beneficial in the management of certain anxiety-related conditions, such as anxiety in medically ill patients, prevention of anxiety following exposure to traumatic stress, and obsessive-compulsive disorder. This paper provides a critical evaluation of the existing evidence base, including unanswered questions that could serve as the focus of further research, and outlines the potential benefits and risks associated with the use of selenium in anxiety disorders.

**Keywords:** selenium; oxidative stress; serotonin; glutamate; anxiety disorders; obsessive-compulsive disorders; post-traumatic stress disorder

## 1. Introduction

Anxiety disorders are among the most common mental disorders worldwide. These disorders are characterized by persistently excessive or inappropriate levels of anxiety, which may occur either spontaneously or in response to specific triggers. The disorders included in this group are generalized anxiety disorder (GAD), panic disorder with or without agoraphobia (PDA), social anxiety disorder (SAD), and specific phobias [1]. It is estimated that around 3–9% of the general population suffers from an anxiety disorder, and another 5–15% fulfill the diagnostic criteria for anxiety-related disorders, such as obsessive-compulsive disorder (OCD) and post-traumatic stress disorder (PTSD) [2–5]. There is a high degree of comorbidity between anxiety disorders, OCD, and PTSD [6,7], which probably reflects common pathogenic mechanisms involving the interaction of genetic vulnerability factors with environmental stressors [8,9]. The pathophysiology of these disorders is complex, involving structural and functional alterations of specific cortical and subcortical brain structures [10–12], dysfunction of specific neurotransmitter systems [13,14], immune-inflammatory pathways [15], oxidative stress [16,17], and impaired neural plasticity [18]. In contemporary guidelines for the pharmacological management of anxiety and related disorders, selective serotonin reuptake inhibitors (SSRIs) and serotonin-norepinephrine reuptake inhibitors (SNRIs) are the recommended first-line treatments [19,20]. Though these groups of drugs are superior to placebos, their effectiveness is often modest [21–23], and only about 40–60% of patients show a clinically significant response to them in short- to medium-term clinical trials [24–26]. Moreover, around 16% of patients who are successfully

treated with SSRIs or SNRIs develop a relapse of anxiety and obsessive-compulsive or post-traumatic stress symptoms when the drug is discontinued after a year of treatment [27], and treatment-emergent adverse effects often lead to non-adherence, particularly at higher doses of medication [28]. Though SSRIs and SNRIs represent a significant advance in the management of anxiety and related disorders, the aforementioned limitations highlight the need for innovative pharmacological treatment approaches that transcend traditional monoamine models of anxiety disorders, and which have a favorable efficacy-to-safety ratio [29–31]. Such treatment methods should ideally target multiple molecular mechanisms known to be involved in the pathogenesis of anxiety disorders [32].

Selenium is a trace element essential for health that is found in several food sources, including animal muscle meats, seafood, grains, and cereals [33]. Its principal biological function is as a component of selenium-containing proteins, or selenoproteins, which are primarily involved in the protection of cells from damage due to oxidative stress. The best-known selenoprotein of this type is glutathione peroxidase (GPx). Other selenoproteins are involved in diverse biological functions such as muscle metabolism, thyroid function, immune-inflammatory responses, and sperm motility [33,34]. More recently, selenoproteins have been identified as playing a significant role in brain development and functioning in mammals [35]. In line with this evidence, some researchers have found evidence of an association between selenium intake and mental health. For example, dietary selenium consumption was negatively associated with symptoms of depression in a study of Brazilian farmers, even after adjusting for demographic and environmental risk factors [36]; similarly, lower plasma selenium was associated with minor depressive symptoms in a sample of post-partum women [37]. Given the high degree of comorbidity and mechanistic overlap between depression and anxiety [38,39], an association between selenium intake or levels and symptoms of anxiety is plausible [40]. There is also recent evidence that selenium may play a key role in preserving normal neural functioning following exposure to stress, which is relevant to the onset and persistence of anxiety disorders [41].

In light of the above considerations, the aim of this review is to examine the evidence base for a link between selenium, selenoproteins, and anxiety disorders at the translational and clinical levels. All included articles will be examined with a focus on the possible use of selenium compounds in the clinical management of anxiety disorders.

## 2. Materials and Methods

The current paper aims to provide a comprehensive review of pre-clinical and clinical research on the relationship between selenium, selenium-containing compounds, and anxiety disorders. Due to the broad nature of the review topic and the difficulty in synthesizing the results of disparate types of research (translational, observational, and interventional), a scoping review methodology was adopted for this review, and it was conducted in adherence with the PRISMA-ScR guidelines [42].

For the purpose of this review, the PubMed, Scopus, and ScienceDirect databases were searched for relevant citations using the search terms "anxiety", "anxiety disorder", "anxiety disorders", "generalized anxiety disorder", "social anxiety disorder", "social phobia", "panic disorder", "agoraphobia", "obsessive-compulsive disorder" and "post-traumatic stress disorder", and their recognized variants, in conjunction with "selenium", "selenoprotein" and "selenoproteins". All studies published up to 20 September 2022, were considered for inclusion. A total of 373 citations were retrieved using this method. Following the removal of duplicate citations, the titles and abstracts of 268 citations were screened for possible inclusion in the review. The following types of original research were considered for inclusion in this review:

- in vitro or animal studies examining the relationship between selenium, selenoproteins, or selenium-containing compounds and experimentally-induced anxiety-like symptoms or behaviors;
- observational studies examining the links between levels of selenium or selenoproteins and symptoms of anxiety or syndromal anxiety disorders in human subjects; and

- interventional studies (clinical trials) examining the therapeutic effect of selenium or selenium-containing compounds in the management of anxiety and related disorders, including obsessive-compulsive disorder (OCD) and post-traumatic disorder (PTSD).

Three citations in languages other than English (two in Russian and one in Hungarian) were retrieved but were not included in the review. Two of these did not fulfill the inclusion criteria for the review. One was a study of mineral toxicity in a rodent model and another was a narrative review of the role of trace elements in mental health. The third was a Russian study of an unspecified selenium-containing dietary product in patients with obesity and cardiovascular disease. This study reported reduced levels of anxiety as a secondary outcome; however, the full text could not be retrieved for translation.

Of the 268 citations screened for inclusion, 58 citations were excluded because they were unrelated to the topic of the review. These included animal or human studies of selenium or selenium-containing compounds in other types of diseases, such as dermatological or endocrine disorders, with no measurement of anxiety-related outcomes. The full text of the remaining 210 articles was examined. Among these, 181 citations were excluded because they represented other publication types, such as general reviews, commentaries, or letters to the editor, that contained no original research or evidence. By searching the references of the remaining twenty-nine papers, one further paper was identified for inclusion. In all, a total of 30 studies were included in this review [43–72].

The included studies were tabulated and were found to belong to the following categories:

(a) studies of selenium, selenoproteins, and selenium-containing compounds in relation to anxiety in animal models (*n* = 16) [43–58];

(b) observational studies measuring dietary selenium intake, levels of serum selenium, or other selenium-related biomarkers in patients with either anxiety disorders or symptoms of anxiety (*n* = 7) [59–65]; and

(c) clinical trials of selenium or selenium-containing supplements in the prevention or management of anxiety and related disorders (*n* = 7), including two studies of selenium and five studies of selenium as one of the components of a nutritional supplement [66–72].

Due to the heterogeneity in patient populations, study objectives, and outcome measures in the clinical trials, a formal meta-analysis was not undertaken for these studies. However, primary and secondary outcomes of efficacy, as well as information related to treatment-emergent adverse events and drop-outs, were tabulated and summarized for each trial.

A flow diagram of the review process is provided in Figure 1.

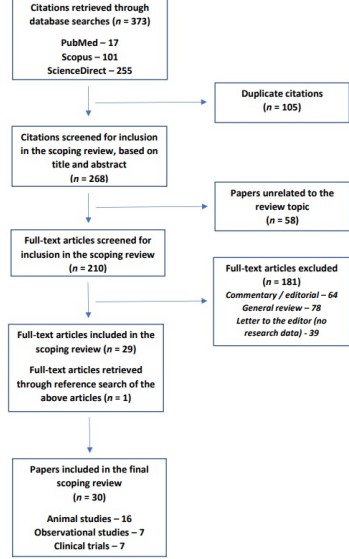

**Figure 1.** PRISMA-ScR flow diagram for the current review.

## 3. Results

### 3.1. Selenium, Selenoproteins, and Selenium-Containing Compounds in Animal Models of Anxiety

Sixteen of the papers included in this review belonged to this category [43–58]. Methodological details and results for each of these studies are summarized in Table 1.

**Table 1.** Animal studies of selenium, selenoproteins, and selenium-containing compounds in relation to models of anxiety.

| Study | Type of Selenium Compound Studied | Animal Species | Experimental Methods | Results—Behavioural | Results—Other |
|---|---|---|---|---|---|
| Peters et al., 2006 [43] | Sodium selenite, high-dose (1 mg/kg) and deficient (0 mg/kg) diets | Male and female mice with targeted disruption of the selenoprotein P gene (SEPP1) and control mice | Observation of anxiety during open-field test and elevated plus maze test; conditioned fear through the pairing of noise with an electric shock to the foot | No difference in anxiety-like behaviors or fear conditioning across SEPP1 genotypes | Reduced spatial learning, short-term plasticity, and long-term potentiation was seen with SEPP1 disruption and selenium-deficient diet |
| Ghisleni et al., 2008 [44] | Diphenyl diselenide (5, 25, and 50 μmol/kg) | Male Wistar rats ($n = 40$, divided into four groups) | Observation of anxiety during the open-field task and the elevated plus maze task | Reduced anxiety, as measured by reduced fecal boli in the open-field task and increased entries and time spent in the open arm of the elevated plus maze, with a 50 μmol/kg dose of diphenyl diselenide only | Anxiolytic effects of diphenyl diselenide abolished by administration of bicuculline (GABA$_A$ antagonist), ketanserin (5HT$_{2A}$ antagonist), or WAY100635 (5HT$_{1A}$ antagonist) |
| Bruning et al., 2009 [45] | *m*-trifluoromethyl-diphenyl diselenide (0.1, 10, and 100 mg/kg) | Female Swiss mice ($n = 40$, divided into four groups) | Observation of anxiety during the elevated plus maze task and light/dark box | Reduced anxiety, as measured by time spent in the illuminated side of a light/dark box and increased entries and time spent in the open arm of an elevated plus maze, with 100 mg/kg *m*-trifluoromethyl diphenyl diselenide | Significant inhibition of cortical MAO-A with 100 mg/kg *m*-trifluoromethyl diphenyl diselenide; anxiolytic effect abolished by administration of WAY100635 (5HT$_{1A}$ antagonist), ritanserin (5HT$_{2A}$ antagonist), or ondansetron (5HT$_3$ antagonist) |
| Gai et al., 2013 [46] | 3-(4-fluorophenyselenyl)-2,5-diphenylselenophene F-DPS) (0.1 mg/kg/day for 1 week) | Male Swiss mice ($n = 48$, divided into four groups) | Induction of anxiety by administration of corticosterone 20 μg/mL of water for 4 weeks | Reduced anxiety, as measured by transition to the dark zone and time spent in the light zone of the light/dark box, with F-DPS | Normalization of ACTH and corticosterone levels, inhibition of cortical MAO-A, and increased 5-HT and glutamate synaptosomal uptake with F-DPS |
| Laureano-Melo et al., 2015 [47] | Sodium selenite, 1 mg/kg, administered to mother rats during pregnancy | Twenty-three Wistar rat offspring, assessed both in childhood and adulthood | Observation of anxiety during the open field test, light–dark test and elevated plus maze test | Reduced anxiety-like behaviors, as measured by increased transitions and time spent in the illuminated side in the light–dark test and an increased time spent in the open arms of the elevated plus maze test during childhood offspring of mothers treated with sodium selenite; the elevated plus maze finding remained significant even when the offspring reached adulthood | Increased serum T$_3$ and T$_4$, reduced hippocampal AChE activity, and reduced hippocampal TPH2 mRNA expression in offspring of mothers treated with sodium selenite |
| Kedzierska et al., 2017 [48] | Sodium selenite, 0.5, 1, and 2 mg/kg | Albino Swiss mice ($n = 40$, divided into four groups) | Observation of anxiety during the adapted elevated plus maze test | Reduced anxiety, as measured by increased time spent in the open arm of the elevated plus maze with all three doses of sodium selenite | This study did not examine any biomarkers related to anxiety |

**Table 1.** *Cont.*

| Study | Type of Selenium Compound Studied | Animal Species | Experimental Methods | Results—Behavioural | Results—Other |
|---|---|---|---|---|---|
| Reis et al., 2017 [49] | 4-phenylselenyl-7-chloroquinoline (4-PSQ), 5–50 mg/kg | Male Swiss mice ($n = 32$, divided into four groups) | Observation of anxiety during the elevated plus maze test, light–dark test, and open field test; induction of anxiety by administration of kainate (15 mg/kg) | Reduced anxiety, as measured by increased time spent in the open arm of the elevated plus maze (25 mg/kg), and reduced transitions (50 mg/kg) and time spent on illuminated size during the light–dark test (25 and 50 mg/kg), with 4-PSQ | A decrease in cortical glutamate uptake, but not glutamate release or $Na^+$, $K^+$-ATPase activity, with 4-PSQ; blockade of kainate-induced anxiety by 4-PSQ (50 mg/kg) |
| Sousa et al., 2018 [50] | α-(phenylselanyl) acetophenone (PSAP), 10 mg/kg | Male Swiss mice ($n = 21$, divided into three groups) | Induction of anxiety by acute restraint stress for 4 h | Reduced anxiety, as measured by a reduced number of marbles buried in the marble burying test and increased entries and time spent in the open arm of the elevated plus maze, with PSAP | A reduction in stress-induced elevations in lipid peroxidation, reactive species, nitrites/nitrates, and corticosterone with PSAP |
| Bampi et al., 2019 [51] | 1-methyl-3-(phenylselanyl)-1$H$-indole (MFSeI), 10 mg/kg | Male Swiss mice ($n = 24$, divided into four groups) | Induction of anxiety by intra-cerebral administration of streptozotocin (0.2 mg/4 µL) | Reduced anxiety, as measured by reduced grooming in the open-field test and increased entries and time spent in the open arm of the elevated plus maze, with MFSeI | A reduction in streptozotocin-induced increases in lipid peroxidation, reactive species, nitrites and AChE activity with MFSeI |
| Casaril et al., 2019 [52] | 3-[(4-chlorophenyl)selanyl]-1-methyl-1$H$-indole (CMI), 1 mg/kg | Male Swiss mice ($n = 32$, divided into four groups) | Induction of anxiety by administration of single-dose LPS (0.83 mg/kg) | Reduced anxiety, as measured by a reduced number of marbles buried in the marble burying test and increased entries and time spent in the open arm of the elevated plus maze, with CMI | A reduction in LPS-induced elevations in IDO, IL-1β, TNF-α with CMI |
| Paltian et al., 2020 [53] | 7-chloro-4-(phenylselanyl) quinoline (4-PSQ), 50 mg/kg | Male Swiss mice ($n = 119$) | Observation of anxiety during the open-field test and elevated plus maze task | Reduced anxiety as measured by increased time spent in the open arm of the elevated plus maze with CMI, but no change in behavior during the open-field test | Anxiolytic effects of 4-PSQ abolished by administration of PTZ ($GABA_A$ antagonist), pindolol (5-HT$_{1A}$ antagonist), or ketanserin (5-HT$_{2A}$ antagonist); 4-PSQ reduced PTZ-induced increase in corticosterone and PTZ-induced decrease in cortical BDNF, CREB, and NF-κB expressions |
| Birmann et al., 2021 [54] | 3,5-dimethyl-1-phenyl-4-(phenylselanyl)-1$H$-pyrazole (SePy) (1 and 10 mg/kg) | Male Swiss mice ($n = 21$, divided into three groups) | Induction of anxiety by acute restraint stress for 2 h | Reduced anxiety, as measured by reduced rearing and grooming in the open-field test, reduced the number of marbles buried in the marble burying test, and increased entries and time spent in the open arm of the elevated plus maze, with both doses of SePy | A reduction in stress-induced elevations in corticosterone, reactive species, lipid peroxidation, and an increase in stress-induced reductions of CAT and SOD with 10 mg/kg SePy |
| Mansouri et al., 2021 [55] | Selenium (specific compound not mentioned in the paper) (100–200 µg/kg) | Male Wistar rats ($n = 32$, divided into four groups) | Induction of anxiety by administration of single-dose LPS (1 mg/kg) | Reduced anxiety, as measured by increased time spent in the open arm of the elevated plus maze test with 200 µg/kg selenium | Increased cortical CAT and SOD in rats treated with Se; greater changes were seen with 200 µg/kg Se |
| Pinz et al., 2021 [56] | 6-((4-fluorophenyl)selanyl)-9$H$-purine (FSP), 1 mg/kg | Male Swiss mice ($n = 28$, divided into four groups) | Induction of anxiety by intracerebral administration of streptozotocin (2 µL of 2.5 mg/mL solution) | Reduced anxiety, as measured by reversal of streptozotocin-induced reduction in dives and open-arm entries during the elevated plus maze test, with FSP | A reduction in streptozotocin-induced an increase in cortical and hippocampal AChE activity and AChE mRNA expression with FSP |

**Table 1.** *Cont.*

| Study | Type of Selenium Compound Studied | Animal Species | Experimental Methods | Results—Behavioural | Results—Other |
|---|---|---|---|---|---|
| Samad et al., 2022 [57] | Sodium selenite (0.175 mg/mL/kg) | Male Wistar rats (*n* = 36, divided into three groups) | Induction of anxiety by exposure to arsenic (2.5 mg/mL/kg for 4 weeks) | Reduced anxiety-like behaviors, as measured by increased time spent in the open arm of the elevated plus maze and light–dark activity tests | Increased levels of GPx, CAT, and SOD in rats treated with Se |
| Situ et al., 2022 [58] | - | Male and female mice with targeted disruption of the selenoprotein W gene (SEPW1) and control mice | Observation of anxiety during the open-field test and elevated plus maze test; conditioned fear through the pairing of noise with an electric shock to the foot | Reduced anxiety-like behavior on the open-field and elevated plus maze tests and impaired fear conditioning in female mice with disruption of SEPW1 | Abnormal hippocampal Nissl bodies neuronal damage and reduced amygdala dendrite spine density in female mice with disruption of SEPW1; no effect of SEPW1 on levels of lipid peroxidation |

Abbreviations: 5-HT$_{1A}$, serotonin type 1A receptor; 5-HT$_{2A}$, serotonin type 2A receptor; 5-HT$_3$, serotonin type 3 receptor; AChE, acetylcholinesterase; BDNF, brain-derived neurotrophic factor; CAT, catalase; CREB, cyclic AMP-response element binding protein; GABA$_A$, gamma-amino butyric acid type A receptor; GPx, glutathione peroxidase; IDO, indoleamine 2,3-dioxygenase; IL-1β, interleukin-1 beta; LPS, lipopolysaccharide; MAO-A, monoamine oxidase type A; NF-κB, nuclear factor kappa B; PTZ, pentylenetetrazol; Se, selenium; SOD, superoxide dismutase; T3, tri-iodothyronine; T4, thyroxine; TNF-α, tumor necrosis factor alpha; TPH2, tryptophan hydroxylase 2.

Fourteen studies examined the effects of selenium or selenium compounds on animal models of anxiety [44–57]. Of these studies, the majority (*n* = 10) used organic selenium-containing compounds as interventions, while two used elemental selenium and two used sodium selenite. All included studies were conducted in mice (*n* = 12) or rats (*n* = 4).

Various procedures were used for the induction of anxiety in experimental animals. In seven studies, anxiety was induced by the administration of a drug or toxin. These included lipopolysaccharide (LPS) (*n* = 2), streptozotocin (*n* = 2), corticosterone, kainate, and arsenic (*n* = 1 each) [46,49,51,52,55–57]. In two studies, anxiety was induced by subjecting animals to acute restraint [50,54]. In the remaining seven studies, no method of symptom provocation was used; instead, anxiety-like behaviors were compared across treated and control animals during the performance of specific experimental tasks, such as the open-field test, the elevated plus maze test, the light–dark box test, and the marble burying test [43–45,47,48,54,58]. All but one of the studies examined adult mice; a single study was conducted in the offspring of rats treated with sodium selenite during pregnancy [47].

In addition to the measurement of anxiety-related behaviors, several of these studies also attempted to measure biomarkers that were associated with the administration of selenium or its compounds, either peripherally or centrally. Peripheral markers that were assayed included peripheral levels of adrenocorticotrophic hormone (ACTH) (*n* = 1), corticosterone (*n* = 2), thyroid hormones (*n* = 1), cytokines (*n* = 1), antioxidant enzymes such as glutathione peroxidase, superoxide dismutase and catalase (*n* = 3), and measures of oxidative stress such as lipid peroxidation, reactive species, and nitrites (*n* = 4). Central measures, estimated through postmortem examination of brain tissue, included the activity of enzymes such as acetylcholinesterase (AChE) (*n* = 3) and monoamine oxidase A (MAO$_A$) (*n* = 1), uptake of neurotransmitters such as glutamate (*n* = 2) and serotonin (*n* = 1), and expression of genes such as brain-derived neurotrophic factor (BDNF), cyclic AMP response element-binding protein (CREB), nuclear factor kappa B (NF-κB), and tryptophan hydoxylase 2 (TPH2) (*n* = 1 each).

Overall, it was consistently observed that both selenium and selenium compounds reduced anxiety-like behaviors during the performance of specific tasks, and also abolished the anxiety-inducing effects of both drugs and toxins. There was evidence of a dose-response relationship in three studies involving organic selenium compounds in adult animals, with only the highest doses exhibiting a meaningful anxiolytic effect [44,45,49]. A single study of selenium also showed evidence of a dose-response effect [55]; however, a study of sodium selenite found that all three doses tested were found to be effective in

reducing anxiety [48]. In studies involving symptom induction by a pharmacological agent, these behavioral effects were largely paralleled by reductions or reversals of the biochemical changes induced by the agent itself; however, none of the studies specifically examined if these biochemical changes were correlated with reductions in anxiety-like behavior.

Among these studies, a single study examined the effect of selenium administration during pregnancy on anxiety in animal offspring [47]. In this study, it was found that sodium selenite supplementation during pregnancy in rats was associated with reduced anxiety-like behavior in offspring both during their childhood and when they reached adulthood. These effects were associated with increases in the levels of thyroid hormones and reductions in brain AChE activity and mRNA expression for the TPH2 gene.

In three of the above studies, all involving organic selenium-containing compounds, researchers examined if the observed anxiolytic effect could be abolished by the administration of pharmacological agents acting on specific neurotransmitter receptors. In these studies, it was found that blockade of gamma-aminobutyric acid type A (GABA$_A$) receptors or serotonin type 1A, 2A, or 3 receptors (5-HT$_{1A}$, 5-HT$_{2A}$, 5-HT$_3$) appeared to abolish the beneficial effects of selenium compounds on anxiety and anxiety-like behaviors [44,45,53].

In addition to these "interventional" studies, two studies examined the effect of the disruption of genes coding for specific selenoproteins by comparing knock-out (KO) animals with wild-type mice. In one of these studies, disruption of the selenoprotein P gene (SEPP1) was associated with impairments in neural plasticity and learning but had no apparent effects on anxiety-like behaviors or fear conditioning [43]. In the second study, disruption of the selenoprotein W gene (SEPW1) was associated with reduced anxiety-like behavior and impaired fear conditioning in female, but not male, mice [58].

### 3.2. Observational Studies of Selenium or Related Biomarkers in Relation to Anxiety in Humans

Seven studies measured dietary selenium intake, peripheral levels of selenium, or specific biomarkers considered to be related to selenium, in human subjects with either anxiety disorders or symptoms of anxiety [59–65]. These studies are summarized in Table 2.

Of the included studies, two studies examined dietary intake of selenium using food frequency questionnaires, one measured serum selenium levels alone, one measured urinary selenium excretion, one measured levels of oxidative stress markers along with serum selenium, and one study examined the expression of specific genes considered to be related to selenium by the researchers. There was significant heterogeneity in patient populations, which included school-age children in the general population ($n = 2$), adults from the general population ($n = 1$), postmenopausal women ($n = 1$), patients with HIV/AIDS ($n = 1$), patients with obsessive-compulsive disorder (OCD) ($n = 1$), and patients with post-traumatic stress disorder ($n = 1$). All the studies involving participants without a psychiatric diagnosis ($n = 5$) used standardized and validated rating scales for the measurement of symptoms of anxiety. Sample sizes were low in studies of patients with a formal psychiatric diagnosis ($n = 28$ for OCD, 8 for PTSD), but were relatively higher ($n = 100$–3846) in the other studies.

The results of this research are mixed, with more consistent results being obtained in clinical samples. Two studies of Chinese children found that serum selenium was significantly associated with both overall and specific symptoms of anxiety [64] but failed to find an association between urinary selenium and symptoms of childhood anxiety [65]. A single study of adults in the general population found no association between a selenium-rich diet (defined as a diet rich in grains and dairy by the researchers) and symptoms of anxiety; however, selenium intake was not specifically quantified in this sample [62]. Likewise, a study of healthy post-menopausal women found no association between serum selenium and anxiety [63]. On the other hand, lower dietary selenium intake was associated with anxiety in patients living with HIV/AIDS [61]. In the two studies involving patients with psychiatric diagnoses, OCD was associated with lower serum selenium, GPx, and catalase, and higher levels of oxidative stress markers [60], while patients with PTSD showed lower levels of expression of two putative selenium-related



genes: thioredoxin reductase (TXR) and superoxide dismutase (SOD). These genes were selected by the researchers as they both encode selenoproteins involved in protection against oxidative stress, and selenium availability may influence peripheral TXR and SOD levels [59]. The study of patients with OCD did not observe any correlation between serum selenium levels and symptom severity.

**Table 2.** Observational studies of selenium or selenium-related biomarkers in relation to anxiety in humans.

| Study | Country of Origin | Study Population and Sample Size | Measure of Anxiety | Parameter Measured | Results |
|---|---|---|---|---|---|
| Zieker et al., 2007 [59] | Germany | Patients with post-traumatic stress disorder (PTSD) following a disaster (*n* = 8) Healthy controls (*n* = 8) | Clinical diagnosis of PTSD according to the Diagnostic and Statistical Manual, 4th edition (DSM-IV) criteria | Expression of selenium-related genes—thioredoxin reductase (*TXR1*) and superoxide dismutase (*SOD1*)—in DNA microarray chips | Significant down-regulation of *TXR1* and *SOD1* in patients with PTSD |
| Ozdemir et al., 2009 [60] | Turkey | Patients with obsessive-compulsive disorder (OCD) (*n* = 28) Age- and sex-matched healthy controls (*n* = 28) | Clinical diagnosis of OCD according to the Diagnostic and Statistical Manual, 4th edition (DSM-IV) criteria | Serum selenium, measured by atomic absorption spectrometry plasma malondialdehyde (MDA), erythrocyte hemolysate glutathione peroxidase (GSH-Px) activity, erythrocyte superoxide dismutase (SOD), and catalase (CAT) activity | Significantly lower selenium GSH-Px and CAT, and higher MDA and SOD, in patients with OCD. No correlation between serum selenium and OCD symptom severity |
| Jamali et al., 2016 [61] | Iran | Patients living with HIV/AIDS (*n* = 100) | Depression, Anxiety, and Stress Scale, 42-item version (DASS-42) | Dietary selenium intake (µg/day) was measured using a 168-item food frequency questionnaire | A significant negative correlation was observed between dietary selenium intake and anxiety score |
| Salehi-Abargouei et al., 2019 [62] | Iran | Adults from the general population (*n* = 3846) | Hospital Anxiety and Depression Scale (HADS) | A selenium-rich diet, measured using a 106-item food frequency questionnaire assessing the daily intake of 57 nutrients | No significant association between a selenium-rich diet and anxiety scores |
| Wieder-Huszla et al., 2020 [63] | Poland | Healthy postmenopausal women (*n* = 102) | State-Trait Anxiety Inventory (STAI) | Serum selenium, measured by absorption spectrometry | No significant association between serum selenium and anxiety score |
| Portnoy et al., 2022 [64] | China | Children from the general population (*n* = 831) | Screen for Child Anxiety Related Emotional Disorders (SCARED) | Serum selenium, measured by atomic absorption spectrophotometry | Lower serum selenium was significantly associated with generalized, social, panic, school-related, and overall anxiety, but not separation anxiety |
| Zhang et al., 2022 [65] | China | Rural children aged 7–11 from the general population (*n* = 831) | Conners' Parent Rating Scale (CPRS) | Urinary selenium, measured by inductively coupled plasma-mass spectrometry | No significant association between urinary selenium and anxiety score |

### 3.3. Clinical Trials of Selenium or Selenium-Containing Supplements for Anxiety

Seven clinical trials examined the efficacy of selenium, or selenium-containing supplements, in the prevention or treatment of patients with anxiety and related disorders [66–72]. These trials are summarized in Table 3.

Among these trials, two were conducted specifically in patients with treatment-resistant obsessive-compulsive disorder (OCD). In the first of these, administration of adjunctive selenium (200 µg/day) was superior to placebo both in terms of OCD symptom reduction and in response rates over a period of 6 weeks; the odds of a clinically meaningful response were approximately six times higher with selenium than with the placebo [70]. In the second, selenium (200 µg/day) was administered as part of a nutraceutical supplement including several other minerals and antioxidants in an open-label design for 20 weeks. In this trial, a significant reduction in OCD symptom scores was also observed, and 23% of patients were classified as responders. It was also found that patients with more severe OCD at baseline were less likely to respond to the selenium-containing supplement [72].

**Table 3.** Clinical trials of selenium or selenium-containing supplements in patients with anxiety and related disorders or symptoms.

| Study | Trial Population and Sample Size | Trial Design | Intervention | Duration | Outcome Measure(s) | Result(s) | Safety and Tolerability Outcomes |
|---|---|---|---|---|---|---|---|
| Gosney et al., 2008 [66] | Nursing home residents aged >60 years, with intact cognition and no major depression or critical medical illness ($n$ = 73) | Randomized controlled trial ($n$ = 36 in active group, $n$ = 37 in placebo group) | A multivitamin and mineral supplement containing selenium (60 μg/tablet), given as four tablets/day (i.e., 240 μg/day selenium) vs. placebo | 8 weeks | Hospital Anxiety and Depression Scale (HADS) | No significant difference between supplement and placebo on HADS anxiety scores | A significantly higher drop-out rate in the supplement group (10/36) than in the placebo group (3/37) |
| Rucklidge et al., 2011 [67] | Adults with attention-deficit hyperactivity disorder (ADHD) exposed to an earthquake ($n$ = 33) | Retrospective analysis of data from two open trials and one randomized controlled trial ($n$ = 16 on supplement and $n$ = 17 not on treatment or placebo) | Micronutrient supplement containing 36 vitamins and minerals, including selenium (26 μg/capsule), given as 15 capsules/day (i.e., 390 μg/day selenium) | 8 weeks | Depression, Anxiety, and Stress Scale, 42-item version (DASS-42) | Significantly lower anxiety scores in the supplement group at 2 weeks post-earthquake (estimated effect size 0.69) | Not reported |
| Voicehovskis et al., 2014 [68] | Military personnel considered at risk for post-traumatic stress disorder (PTSD) ($n$ = 97) | Controlled clinical trial; randomization not mentioned ($n$ = 64 for selenium and $n$ = 33 for placebo) | Selenium (200 μg/day) vs. placebo | 6 months | Changes in PTSD Symptom Checklist—Military Version (PCL-M) Measurement of malondialde-hyde (MDA) in a subset ($n$ = 62) of the sample | A reduction of 46.03% in those screening positive for PTSD on the PCL-M in the selenium group; a significant reduction in MDA with selenium compared to the placebo | Not reported |
| Kaplan et al., 2015 [69] | Adults aged 23–66 exposed to a flood ($n$ = 56) | Randomized clinical trial ($n$ = 17 in vitamin D group, $n$ = 21 in few-nutrients group, $n$ = 18 in broad-spectrum mineral/vitamin group) | Broad-spectrum mineral/vitamin (BSMV) supplement including selenium (45.2 μg/capsule), given as four capsules/day (i.e., ≈ 180 μg/day selenium) vs. vitamin D (1000 IU/day) and B-complex supplement (1 capsule/day) | 6 weeks | Depression, Anxiety, and Stress Scale, 42-item version (DASS-42) | A significant reduction in anxiety in the BSMV and B-complex groups compared to vitamin D (estimated effect size 0.89); no significant difference between the BSMW and B-complex groups | Minor adverse events (headache, nausea, rash) were comparable across groups; no significant difference in drop-out rates between groups |
| Sayyah et al., 2018 [70] | Patients with treatment-resistant obsessive-compulsive disorder ($n$ = 32) | Randomized controlled trial ($n$ = 16 in each group) | Selenium (200 μg/day) vs. placebo | 6 weeks | The mean reduction in the Yale-Brown Obsessive-Compulsive Scale (Y-BOCS) score response, defined as ≥25% reduction in Y-BOCS score | The mean reduction in Y-BOCS was significantly greater in selenium; response rates were 43.7% in the selenium group and 7.1% in the placebo group | Adverse events (sedation, constipation, nausea, tremor, sexual dysfunction) comparable between selenium and placebo; no significant difference in drop-out rates between groups |
| Raygan et al., 2019 [71] | Patients with diabetes mellitus (type II) and coronary heart disease ($n$ = 54) | Randomized controlled trial ($n$ = 27 in each group) | Combined selenium (200 μg/day) and probiotic ($8 \times 10^9$ CFU/day) vs. placebo | 12 weeks | Beck Anxiety Inventory (BAI) | A significant reduction in BAI in the supplement group compared to the placebo group (mean difference −1.46) | No adverse effects were reported by participants; no significant difference in drop-out rates between groups |

**Table 3.** *Cont.*

| Study | Trial Population and Sample Size | Trial Design | Intervention | Duration | Outcome Measure(s) | Result(s) | Safety and Tolerability Outcomes |
|---|---|---|---|---|---|---|---|
| Sarris et al., 2021 [72] | Patients with treatment-resistant obsessive-compulsive disorder (*n* = 28) | Open-label trial | A supplement containing selenium, zinc, magnesium, pyridoxal 5′-phosphate, N-acetyl cysteine, and L-theanine | 20 weeks | A mean reduction in the Yale-Brown Obsessive-Compulsive Scale (Y-BOCS) score response, defined as ≥35% reduction in Y-BOCS score | A significant mean reduction (7.13 points) in Y-BOCS; 23% of patients were classified as responders | The treatment was reported as well-tolerated overall; no treatment drop-outs |

Note: The specific selenium compound used in each of these trials was not reported in the paper; doses were provided in terms of micrograms of elemental selenium.

Two trials examined the efficacy of selenium or selenium-containing supplements in the treatment of symptoms of anxiety in specific clinical populations. In a trial involving 54 patients with comorbid type II diabetes mellitus and coronary artery disease, a combination of selenium (200 μg/day) and a probiotic was superior to a placebo in reducing self-reported symptoms of anxiety [71]. On the other hand, a trial of a multi-vitamin and mineral supplement containing selenium (240 μg/day) in geriatric nursing home residents found no significant effect of this treatment on clinician-rated symptoms of anxiety, though it was superior to the placebo in reducing symptoms of depression [66].

The remaining three trials examined the efficacy of selenium or selenium-containing supplements in the prevention of anxiety-related symptoms. Two of these were conducted on civilians, and one on military personnel. In the first of the civilian studies, data on adults with a diagnosis of attention-deficit hyperactivity disorder enrolled in three ongoing trials of a selenium-containing dietary supplement (selenium 390 μg/day); they were retrospectively analyzed following exposure to an earthquake. It was found that patients who were receiving the supplement had significantly lower levels of anxiety compared to those on the placebo or no treatment, with a medium effect size (0.69) [67]. In an attempt to replicate this work, a randomized clinical trial involving three treatment arms—vitamin D, vitamin B complex supplementation, and a broad-spectrum mineral-vitamin (BSMV) supplement containing selenium (≈180 μg/day)—was conducted in adults without a psychiatric diagnosis exposed to a flood. In this trial, both the BSMV supplement and the vitamin B complex supplement were superior to vitamin D in reducing symptoms of anxiety; however, there was no significant superiority of the BSMV supplement over vitamin B complex (effect size 0.12). The estimated effect size in this trial (for BSMV vs. vitamin D) was somewhat larger (0.88) [69]. Finally, in a trial involving 97 Latvian army personnel considered to be at risk of post-traumatic stress disorder (PTSD), treatment with 200 μg/day of selenium for 6 months was associated with a lower screen-positive rate for PTSD; selenium treatment was also associated with lower levels of malondialdehyde, a marker of lipid peroxidation [68].

Safety data were presented in five of the seven trials. Adverse events reported with selenium or selenium-containing supplements were generally mild in severity and included headache, nausea, and constipation [69,70]. More significant adverse events included sexual dysfunction, sedation, and tremor; however, as patients reporting the latter adverse events were also taking antidepressants, it is not clear whether selenium was responsible for their emergence [70]. In general, treatment drop-outs due to adverse effects (in the trials involving control groups) were not increased in selenium-treated patients [69–72]. The only exception to these general observations was a trial involving elderly patients (a mean age of 82 years), where drop-outs from treatment were higher in the treatment group (10 of 37) than in the placebo group (3 of 36); this difference was statistically significant ($p$ = 0.035, Fisher's exact test) [66].

As all the included studies involved small samples, and there was wide variation both in terms of treatment interventions (i.e., selenium vs. selenium-containing supplements) and outcomes (anxiety, OCD symptoms, and PTSD symptoms), it was not possible to conduct a formal meta-analysis or to systematically assess publication bias. However, some of the study methods adopted (retrospective analysis of pooled open-label and randomized data, *n* = 1; method of randomization not described, *n* = 1; no placebo or control group, *n* = 1) could potentially lead to a bias towards positive or favorable study results.

## 4. Discussion

Selenium is an essential micronutrient involved in several biological processes, most specifically as a component of various selenoproteins. Recent evidence suggests that the role of selenium and selenoproteins in brain functioning extends beyond protection against oxidative stress, and may involve protective effects or functional interactions with dopaminergic, cholinergic, and GABAergic pathways, as well as the effect of selenoprotein P on specific post-synaptic receptors [73]. There is also evidence that selenium can alter the functioning of the gut–brain axis [74] and may be involved in the regulation of the stress response through interaction between glucocorticoids and selenoproteins [41]. This evidence has led to an interest in the use of selenium or its compounds in the management of common mental disorders, namely, depression and anxiety disorders. An earlier review of the literature suggested that while selenium was a theoretically useful adjunctive approach to the management of depression and anxiety, there was no firm clinical evidence of its efficacy [75]. Subsequent research has shown that selenium supplementation, at a dose of 30–200 μg/day, appears to be effective in the prevention and treatment of depression in diverse clinical populations, including pregnant women [76], women with polycystic ovarian disease [77], and adults with comorbid obesity and depressive symptoms [78]; in one of these trials, selenium appeared to be effective in reducing scores on a composite measure of anxiety and depression, even though anxiety sub-scores were not analyzed separately [77]. Selenium supplementation was well-tolerated in these trials, with few significant adverse effects or elevated drop-out rates. These findings provide a rationale for the possible use of selenium, either alone or as part of a nutritional supplement, in the management of anxiety and related disorders.

The data reviewed from animal studies in this paper provides further support for this suggestion, suggesting that both selenium and selenium-containing organic compounds were effective in reducing anxiety-like behaviors in rodents. This research also identified some of the possible molecular mechanisms underlying the anxiolytic effect of selenium, involving reductions in oxidative stress [50,51,54,55,57], reductions in the levels of pro-inflammatory cytokines [52], normalization of stress-induced changes in hypothalamic-pituitary-adrenal axis functioning [46,54], interactions with neurotransmitters such as serotonin, glutamate, and GABA [44–47,49,50,53], and alterations in thyroid functioning [47]. However, these results should be interpreted with caution for the following reasons. First, while some of the animal models used in this research—such as observation during task performance, exposure to restraint stress, and administration of glucocorticoids—may be relevant to the onset or persistence of anxiety disorders in humans, others, such as the administration of toxins, cannot be interpreted in this manner. Second, the majority of such studies involved complex organic molecules containing selenium; it is not clear to what extent the presence of selenium in these compounds—as opposed to other aspects of their chemical structure—was associated with a specific anxiolytic effect. Third, most of the studies identified in this review were of short duration, whereas anxiety disorders in humans tend to be chronic in nature. Fourth, no meaningful information on safety in humans could be derived from these short-term trials. Despite these limitations, the results of the animal models suggest that selenium and selenium-containing compounds may exert an anxiolytic effect through multiple mechanisms, giving them a theoretical advantage over existing drugs that act through a limited number of neurotransmitter pathways [32]. A further finding of interest in these studies, though requiring replication, is the possible

effect of selenoprotein W on fear conditioning, suggesting that the expression of this protein may underlie differences in individual vulnerability to anxiety and related disorders [58].

There are relatively few observational studies examining the links between selenium intake, serum levels of selenium, or selenium-related biomarkers and anxiety in humans, and the results of these studies do not suggest a consistent association between selenium levels and anxiety. At least a portion of this variation could be due to differences in study methodology—studies measuring dietary selenium intake through questionnaires or urinary selenium were more likely to yield negative results [62,65], as were studies involving healthy individuals with no known medical or psychiatric diagnosis [62,63]. Other methodological factors likely to be involved include the nature of the questionnaires used for the assessment of anxiety. For example, a study making use of a multi-dimensional assessment of different types of anxiety was able to establish a tentative link between serum selenium and some of the dimensions of anxiety [64]. Finally, an important confounding factor in research of this sort is baseline selenium intake, which varies widely across countries depending on dietary practices and soil selenium levels [79,80]. The possible associations between selenium, oxidative stress markers, and OCD [60], and between selenium-related gene expression and PTSD [61] require replication, and further studies of selenium and related biomarkers in other anxiety disorders, such as generalized anxiety disorder, panic disorder, and social anxiety disorder, are warranted. Few definitive conclusions can be drawn from the available observational data.

Based on data from controlled clinical trials, there is replicated evidence for a possible role of selenium in the treatment of OCD, and more specifically OCD that is resistant to standard modes of treatment, such as serotonin reuptake inhibitors and behavior therapy [70,72]. However, even in this case, only a minority of patients were considered to have shown a satisfactory response to treatment, and this number decreased when a more stringent definition of response (35% reduction in OCD symptom scores, as opposed to 25%) was used [72]. Given that this treatment was well-tolerated in both studies, this approach merits further exploration, particularly replication in larger samples.

The evidence for a combination of selenium and probiotics in the management of anxiety in a medically ill sample, though based on a study of good methodological quality, cannot easily be generalized to patients with anxiety disorders in the community. This is because the pathophysiology of anxiety in patients with cardiac or metabolic disorders may be distinct from that observed in patients with anxiety and no medical comorbidities [81,82]. Likewise, evidence on the role of selenium-containing supplements in preventing anxiety in individuals exposed to severe stress, though promising, cannot be easily translated into clinical recommendations due to the methodological limitations of the existing studies. Moreover, the possibility of a bias towards the publication of positive results, which is high in studies related to psychopharmacology, should always be borne in mind [83], particularly when proprietary products are used [84], which was the case in two of the "preventive" trials [67,69].

Concerns related to the safety of selenium, either alone or as part of a supplement, also require consideration in the context of controlled clinical trials. Of the seven trials, only four reported data on adverse events, and five provided information on drop-out rates. Though reported adverse effects tended to be mild, this information was derived from short- to medium-term clinical trials, and it is not known if more severe adverse events would emerge if selenium-containing supplements were administered for a longer period of time [85]. Data from a single trial suggest that elderly patients may be particularly likely to discontinue this form of treatment prematurely, though it was not specifically mentioned if this was due to adverse events [66].

Certain limitations of the current review require elaboration. First, though animal studies suggest that selenium or selenium-containing organic compounds have anxiolytic properties, these findings cannot be directly "translated" to human patients with a clinical diagnosis of anxiety disorder, and any molecular "lead" obtained from this research should be carefully evaluated for both safety and efficacy. Second, there is a paucity of



observational studies on the link between selenium and syndromal anxiety disorders in humans; it is not possible to conclude that any significant relationship exists based on the available data, except possibly in patients with OCD and in children. Third, other than the trials involving OCD patients, no controlled clinical trial has evaluated the efficacy and safety of selenium or selenium-containing supplements in other well-defined anxiety disorders, such as generalized or social anxiety disorder. Fourth, many of the human clinical trials involved included selenium as part of a supplement containing several other vitamins and minerals; it cannot be stated for certainty that selenium was responsible for the clinical benefits observed [67,69,71,72]. Fifth, it is not known if selenium is effective in all patients with anxiety or only in a subset with specific biological changes, such as low serum selenium or elevated markers of oxidative stress [75]. Sixth, a single trial of potential relevance could not be retrieved due to linguistic and electronic access barriers. Based on the abstract, it is possible that the inclusion of this study could have strengthened the evidence for an anxiolytic effect of selenium in medically ill subjects [86]. Finally, there are risks inherent in selenium supplementation, particularly in those with a high dietary intake of this micronutrient [87], and a narrow therapeutic index may limit the real-world efficacy of selenium-based treatments [85].

## 5. Conclusions

The available translational and clinical evidence is consistent with a possible role of selenium supplementation in the management of anxiety disorders. However, variations in the quality and quantity of this evidence do not permit a more definitive conclusion to be drawn. However, the existing literature does provide a clear delineation of research questions that need to be answered before selenium can be used to treat anxiety disorders in clinical practice. First, the basic neurobiological mechanisms involved in the effects of selenium on anxiety need to be elucidated in greater detail, through the use of in vitro and animal models examining this element in relation to a neurotransmitter, endocrine, immune-inflammatory, and oxidative stress pathways. This would also include a clearer characterization of the receptor-binding profiles of organic selenium compounds and whether their effects are critically dependent on their selenium content. Second, observational studies of serum selenium levels should be carried out both in large general population samples (to examine the relationships between normal and subnormal selenium levels with subsyndromal levels of anxiety) and in patients with a clinically diagnosed anxiety disorder (to examine relationships between selenium levels and specific diagnoses, symptom dimensions, or symptom severity). The incorporation of biomarkers related to the physiological properties of selenium, such as measures of oxidative stress, would enhance the quality of such studies and allow for a better delineation of underlying biological mechanisms. Finally, controlled clinical trials should focus on the use of supplemental selenium rather than a wide-ranging multivitamin and mineral supplementation. Such trials should incorporate measurement of baseline and post-treatment selenium levels to identify a possible relationship between baseline selenium and treatment response, and should also report data on adverse effects and treatment tolerability with greater rigor. Further issues of clinical relevance, such as long-term safety and efficacy and the identification of biomarkers of response (endocrine, oxidative stress-related, or neurochemical), would depend upon the demonstration of safety and efficacy in short-term, randomized controlled trials of selenium in anxiety disorders.

**Funding:** This research received no external funding.

**Informed Consent Statement:** Not applicable.

**Conflicts of Interest:** The author declares no conflict of interest.

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
