# Peer review of "Selenium and Its Compounds in the Treatment of Anxiety and Related Disorders: A Scoping Review of Translational and Clinical Research"

_futurepharmacol, doi:10.3390/futurepharmacol2040037_

Round 1

Reviewer 1 Report

The manuscript in question provides a critical evaluation of evidence on the potential role of selenium and its compounds in the management of anxiety and related disorders. This is an important contribution to the field as it presents a step forward to the understanding of the place of selenium in today’s rampant diseases, such as anxiety and many others.

In addition, the article is written in good English, thoughts and ideas are organized in a naturally flowing text making it an interesting and exciting read.

However, there is a certain, minor issue that need to be resolved.

In the Materials and methods section, it should be made clear how the search was performed in terms of the language/s used.

It should be stated whether only articles written in English were included in the study, or if the study included data published in other languages.

If non-English articles were indeed used during the search and writing process, the potential cultural (language-related) bias should be properly addressed, possibly in the Limitations section (lines 389 and so on).

Author Response

I thank the reviewer for their thoughtful review of my manuscript.

I agree with the reviewer that exclusion of papers written in other languages may lead to a bias in the presentation of evidence.

The relevant searches were repeated without any language filter and the following revisions have been made:

Methodology (lines 105-111)

"Three citations in languages other than English (two in Russian and one in Hungarian) were retrieved, but were not included in the review. Two of these did not fulfill the inclusion criteria for the review: one was a study of mineral toxicity in a rodent model, another was a narrative review of the role trace elements in mental health. The third was a Russian study of an unspecified selenium-containing dietary product in patients with obesity and cardiovascular disease. This study reported reduced levels of anxiety as a secondary outcome; however, the full text could not be retrieved for translation."

Limitations (lines 429-433):

"Sixth, a single trial of potential relevance could not be retrieved due to linguistic and electronic access barriers; based on the abstract, it is possible that the inclusion of this study could have strengthened the evidence for an anxiolytic effect of selenium in medically ill subjects [86]."

Reviewer 2 Report

I found this review article well written and organized. The method section clearly explains how the research was approached and the results are clearly presented and explained. I believe this is a great contribution to the field of pharmacological approaches to treat anxiety-related disorders and provide a good overview of past evidence and possible future research directions. As minor revisions, I would suggest the author to double check the abbreviations (e.g., OCD) and some repetitions (e.g., line 30 included-include) in the same sentences. 

Author Response

I thank the reviewer for their thoughtful review of my manuscript.

The paper has been checked for errors in word selection and use of abbreviations and the following corrections have been made:

Lines 29-32: "The disorders included in this group are generalized anxiety disorder (GAD), panic disorder with or without agoraphobia (PDA), social anxiety disorder (SAD) and specific phobias [1]."

Lines 100-104: "interventional studies (clinical trials) examining the therapeutic effect of selenium or selenium-containing compounds in the management of anxiety and related disorders, including obsessive-compulsive disorder (OCD) and post-traumatic disorder (PTSD)"

Reviewer 3 Report

The manuscript from  Rajkumar RP is well-written and organized. The author made a great overview of the effects of Se on anxiety. Before acceptance, I have a few questions/suggestions.

Peters et al., 2006 [43]. Please identify the selenium chemical form.

Kedzierska et al., 2017 [48]. Why the results were not acessed?

Mansouri et al., 2021 [55]. Please identify the selenium chemical form.

Samad et al., 2022 [57]. Please identify the selenium chemical form.

Table 2. Türkiye. Please change it to Turkey

"...of two putative selenium-related genes: thioredoxin reductase and superoxide dismutase [59]." How is SOD gene related with Se?

Table 3. Please identify the selenium chemical form in the studies.

Author Response

I thank the reviewer for their thoughtful review of my manuscript.

The following corrections have been made based on the review report:

1. Peters et al., 2006 [43]. Please identify the selenium chemical form.

Response: This has been corrected to mention the chemical form (sodium selenite) used in this study.

2. Kedzierska et al., 2017 [48]. Why the results were not acessed?

Response: I apologize for the lack of clarity in the text. The results were accessed. What I meant to convey was that this study examined only behavioural measures of anxiety and did not measure any biomarker levels. This has been mentioned in the revised table: "This study did not examine any biomarkers related to anxiety."

3. Mansouri et al., 2021 [55]. Please identify the selenium chemical form.

Response: I have checked the original article text. However, the authors did not mention the specific chemical form used. This has been mentioned in the table: "Selenium (specific compound not mentioned in the paper) (100-200 µg/kg)"

4. Samad et al., 2022 [57]. Please identify the selenium chemical form.

Response: I apologize for this oversight. It has been corrected to "Sodium selenite (0.175 mg/ml/kg)".

5. Table 2. Türkiye. Please change it to Turkey

Response: The spelling has been corrected to "Turkey".

6. "...of two putative selenium-related genes: thioredoxin reductase and superoxide dismutase [59]." How is SOD gene related with Se?

Response: The text has been revised to include the researchers' rationale: "These genes were selected by the researchers as they both encode selenoproteins involved in protection against oxidative stress, and selenium availability may influence peripheral TXR and SOD levels."

7. Table 3. Please identify the selenium chemical form in the studies.

Response: I agree with the reviewer that this is an important detail. However, on reviewing each of the included studies, none of them have mentioned the specific compound used; instead, all of them have provided a dose in terms of equivalent elemental selenium. Hence, a footnote has been added to the table explaining this limitation, as follows:

"Note: The specific selenium compound used in each of these trials was not reported in the paper; doses were provided in terms of micrograms of elemental selenium."